Peer**J**

# Oxygen availability is a major factor in determining the composition of microbial communities involved in methane oxidation

Maria E. Hernandez[1,4], David A.C. Beck[1,3], Mary E. Lidstrom[1,2] and Ludmila Chistoserdova[1]

[1] Department of Chemical Engineering, University of Washington, Seattle, USA
[2] Department of Microbiology, University of Washington, Seattle, USA
[3] eScience Institute, University of Washington, Seattle, USA
[4] Biotechnological Management of Resources Network, Institute of Ecology, A.C. Xalapa, Veracruz, Mexico

Corresponding author
Ludmila Chistoserdova, milachis@u.washington.edu

## ABSTRACT

We have previously observed that methane supplied to lake sediment microbial communities as a substrate not only causes a response by *bona fide* methanotrophic bacteria, but also by non-methane-oxidizing bacteria, especially by members of the family *Methylophilaceae*. This result suggested that methane oxidation in this environment likely involves communities composed of different functional guilds, rather than a single type of microbe. To obtain further support for this concept and to obtain further insights into the factors that may define such partnerships, we carried out microcosm incubations with sediment samples from Lake Washington at five different oxygen tensions, while methane was supplied at the same concentration in each. Community composition was determined through 16S rRNA gene amplicon sequencing after 10 and 16 weeks of incubation. We demonstrate that, in support of our prior observations, the methane-consuming communities were represented by two major types: the methanotrophs of the family *Methylococcaceae* and by non-methanotrophic methylotrophs of the family *Methylophilaceae*. However, different species persisted under different oxygen tensions. At high initial oxygen tensions (150 to 225 µM) the major players were, respectively, species of the genera *Methylosarcina* and *Methylophilus*, while at low initial oxygen tensions (15 to 75 µM) the major players were *Methylobacter* and *Methylotenera*. These data suggest that oxygen availability is at least one major factor determining specific partnerships in methane oxidation. The data also suggest that speciation within *Methylococcaceae* and *Methylophilaceae* may be driven by niche adaptation tailored toward specific placements within the oxygen gradient.

## INTRODUCTION

Methanotrophy is a well-characterized mode of microbial metabolism that supports microbial growth on methane (*Trotsenko & Murrell, 2008*). Methanotrophs are important players in the methane cycle, and, more generally, in the carbon cycle on Earth (*Singh et al., 2010*; *Nisbet, Dlugokencky & Bousquet, 2014*). A variety of methane-oxidizing microbes have been characterized in pure cultures (most prominently the organisms belonging to Proteobacteria) but also more recently, organisms classified as Verrucomicrobia, the latter so far only found in extreme environments (*Chistoserdova & Lidstrom, 2013*). While methanotrophy can be carried out by single species, it has been noted that methanotrophs in environmental samples are often associated with specific non-methanotrophic bacteria (*Jensen et al., 2008*; *Redmond, Valentine & Sessions, 2010*; *He et al., 2012*; *Dubinsky et al., 2013*; *Rivers et al., 2013*), suggesting some type of cooperation (*Beck et al., 2013*; *Van der Ha et al., 2013*). We have previously tested for the possibility of such cooperative behavior by analyzing the compositions of microcosms originating from Lake Washington sediment which was exposed to methane as the only carbon source and observed a prominent presence of satellite bacteria (*Oshkin et al., 2014*). Among the most persistent satellites, we identified members of the families *Methylophilaceae* and *Flavobacteriaceae* (*Oshkin et al., 2014*). These observations further suggested a novel metabolic framework for methane oxidation as carried out by communities of different metabolic guilds, rather than methanotrophs alone. However, additional experimental support was necessary in order to shift the accepted paradigms of methane oxidation (*Trotsenko & Murrell, 2008*). As methane oxidation in environments such as lake sediments takes place over steep counter gradients of methane and oxygen (*Auman et al., 2000*), the focus of this study was on investigating the effect of oxygen availability on bacterial community structure in microcosms enriched with methane as a substrate.

## MATERIAL AND METHODS

### Sample collection and experimental setup

Samples of Lake Washington sediment were collected on July 15, 2013 (*Oshkin et al., 2014*). A 50 ml frozen sediment sample containing 10% of dimethyl sulfoxide (a cryoprotective agent) was thawed on ice, mixed and used as an inoculum. Five ml aliquots of sediment slurry were placed into 250 ml vials and diluted with 50 ml of nitrate mineral salts (NMS) medium (*Dedysh & Dunfield, 2014*; 0.5 X strength), vials were sealed with rubber stoppers and flushed with $N_2$ for 2 min (flow rate 400 ml/min), and the excess volume of $N_2$ was removed by a syringe to equalize the pressure. Five different atmospheres were created in the headspaces by adding different volumes of ambient air, as follows: 5%, 15%, 25%, 50% or 75% of the headspace (V/V). All headspaces received 25% (V/V) of methane. Before adding the air and the methane, the respective volumes of $N_2$ were removed from the vials. These initial oxygen tensions correspond to, respectively, approximately 15, 45, 75, 150, and 225 µM of dissolved oxygen. Three replicate microcosms for each oxygen tension were incubated in a shaker (250 RPM) at 18 °C. The headspace gas composition was recreated daily, as above. After three weeks of incubation, the microcosms were transferred into

new medium, with 10-fold dilutions, similarly to the procedure described previously (*Oshkin et al., 2014*). Such transfers were then repeated through week 16.

## Oxygen and methane measurements

Oxygen and methane concentrations in the headspace were measured using a GC2014 gas chromatograph (Shimadzu Instruments, Pleasanton, California, USA) as described by *Oshkin et al. (2014)*.

## 16S rRNA gene amplicon sequencing

Cell biomass was collected at weeks 10 and 16. DNA was isolated using the FastDNA SPIN KIT for Soil (MP Biomedicals, Burlingame, California, USA) and submitted to MR DNA service facility (www.mrdnalab.com; Shallowater, TX, USA). PCR primers 27F/519r with barcode on the forward primer were used in a 30 cycle PCR using the HotStarTaq Plus Master Mix Kit (Qiagen, Valencia, California, USA) under the following conditions: 94 °C for 3 min, followed by 28 cycles of 94 °C for 30 s, 53 °C for 40 s and 72 °C for 1 min, after which a final elongation step at 72 °C for 5 min was performed. After amplification, PCR products were checked in a 2% agarose gel to determine the success of amplification and the relative intensity of bands. Multiple individual samples were pooled together in a way that each sample was represented equally, for multiplexing. Pooled samples were purified using calibrated Ampure XP beads. Then the pooled and purified PCR products were used to prepare DNA libraries following the manufacturer's instructions. Sequencing was performed on a MiSeq instrument following the manufacturer's guidelines. Sequence data were processed using a proprietary MR DNA analysis pipeline in which sequences barcodes were removed, then sequences <150 bp or with ambiguous base calls were removed, sequences were denoised and chimera sequences were removed. The pairs of sequences were joined, resulting in sequences between 490 and 492 nucleotides. The data have been archived with the NCBI (Bioproject PRJNA274703, http://www.ncbi.nlm.nih.gov/bioproject/?term=PRJNA274703).

## Bioinformatics

The UPARSE method was used for sequence processing and OTU clustering with USEARCH version 7.0.1001 (*Edgar, 2013*). Clustering was performed at 95% and chimeras were identified against the ChimeraSlayer reference database in the Broad Microbiome Utilities version r20110519 obtained from the UCHIME distribution (*Edgar et al., 2011*). For each OTU, a representative sequence was selected using the method of *Edgar (2013)*, and taxonomic assignments were made using the RDP Classifier from the Ribosomal Database Project downloaded on October 22, 2013 (*Wang et al., 2007*). The samples were scaled so that the numbers of reads in each sample were equal. Hierarchical clustering of samples and OTUs was performed using the percentage of reads per OTU for the most abundant taxa, i.e., greater than 1.0% population in at least one sample. Bray-Curtis distances and Shannon indices were calculated and multivariate analyses were carried out using the *vegan* library version 2.0-10 (*Oksanen et al., 2014*) in R version 3.0.2

 

(http://www.R-project.org/). The processing and analysis code has been made available (DOI 10.5281/zenodo.13190).

Genome–genome comparisons were carried out using the Phylogenetic Profilers tool that is part of the Integrated Microbial Genomes database (IMG/JGI; https://img.jgi.doe.gov). Reciprocal searches were performed to determine all the genes present in both *Methylobacter* genomes but absent in the *Methylosarcina* genome and vice versa, and searches were performed to determine all the genes present in both *Methylophilus* genomes but in none of the *Methylotenera* genomes and vice versa, using 30% protein sequence cutoff.

## RESULTS AND DISCUSSION

Previously, we had followed short-term community dynamics in microcosms of Lake Washington sediment, under an atmosphere of methane and two oxygen tension regimens, 'high' and 'low' (*Oshkin et al., 2014*). However, with the experimental design utilized, the communities were limited by oxygen in both conditions for extended periods of time. Under both regimens, we observed rapid loss of community complexity and establishment of stable communities dominated by *Methylobacter*, a gammaproteobacterial methanotroph, and by members of *Methylophilaceae* (*Methylotenera* or *Methylophilus*), non-methanotrophic methylotrophs within Betaproteobacteria. We have also noted persistent presence of certain non-methylotrophic heterotrophs, such as *Flavobacteriaceae* (*Oshkin et al., 2014*).

In the research described here, one of our goals was to test for the reproducibility of microcosm trajectories, adding a few modifications to the experimental design (such as a slightly modified medium, slightly higher temperature, and most notably a different sequencing technology) and to confirm the persistence of specific bacterial taxa in such microcosms, under the selective pressure of methane. Our second goal was to test a broader range of oxygen tensions, with a stricter control over the oxygen concentration in the headspace. We employed five discrete initial oxygen tensions, calculated to correspond to approximately 225, 150, 75, 45 and 15 μM dissolved oxygen, mimicking the oxygen gradient between 0 and 5 mm in the native sediment of Lake Washington where most of the methane oxidizing activity takes place (*Auman et al., 2000*). With the headspace compositions recreated daily, only the communities exposed to 150 and 225 μM initial dissolved oxygen remained oxygenated (starting with week 7 for the 225 μM oxygen community and week 8 for the 150 μM community; Fig. S1). Communities exposed to lower initial oxygen tensions depleted oxygen before the next addition (Fig. S1). Typical rates of methane and oxygen consumption in the established communities are shown in Fig. 1.

The community composition was measured after ten and sixteen weeks of incubation, in three replicated microcosms for each oxygen tension (Table S1). Illumina-based analysis of microcosm communities uncovered that they were simple communities (20 to 40 operational taxonomic units (OTUs) per microcosm; Fig. S2), and they contained two dominant bacterial guilds, methanotrophs of the family *Methylococcaceae* and methylotrophs of the family *Methylophilaceae*. In most of the microcosms (66.6%), sequence reads ascribed to these two functional guilds made up over 90% of all reads

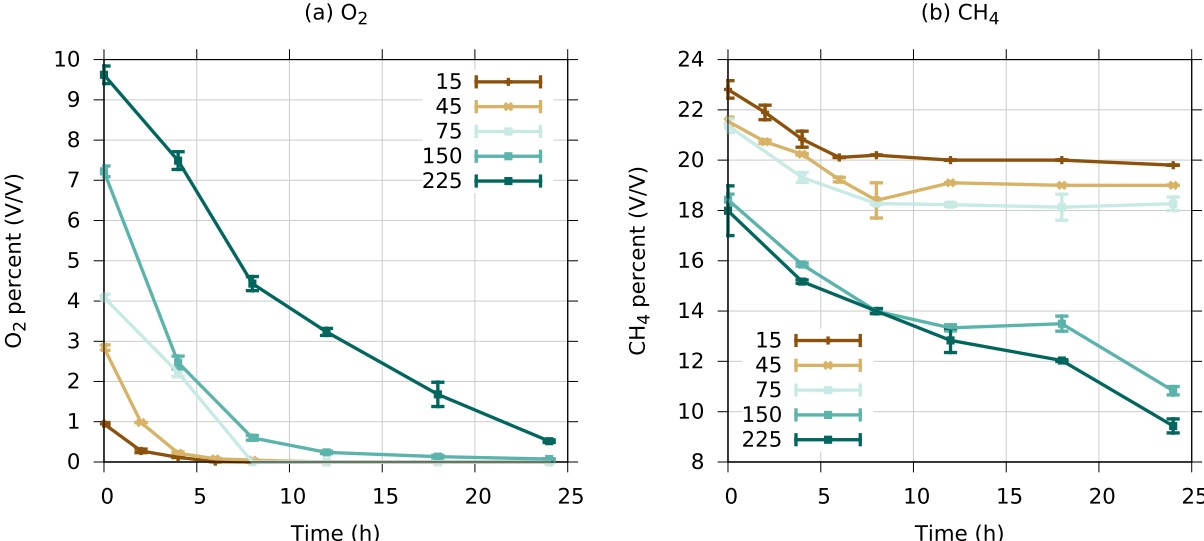

**Figure 1 Typical dynamics of oxygen (A) and methane (B) consumption in low complexity microcosms, over the course of 24 h.** For this experiment, six additional replicates were prepared for each microcosm at week 16, and these were allowed to incubate for 48 h, with the atmospheres recreated at the 24-h point. At the 48-h point, the atmospheres were recreated again, and measurements were taken every two (15 and 45 μM treatments) or four (75 to 225 mM treatments) hours. Bars indicate standard error across the replicates.

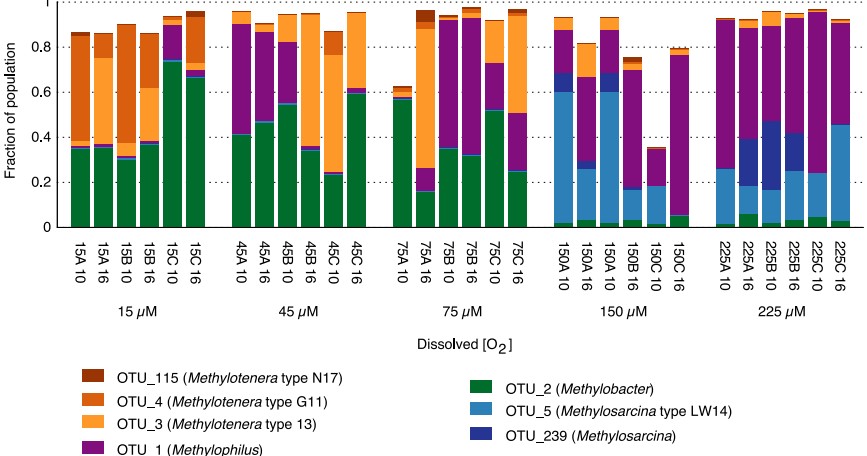

**Figure 2 Relative abundance of *Methylococcaceae* and *Methylophilaceae* in methane-fed microcosms.** Samples were ordered from the lowest to the highest concentration of oxygen. Sample designations include oxygen tension, followed by the alphabetical name of a replicate and by the week of sampling.

(Fig. 2). These data are in agreement with the data from our prior study, in which similar community structures were observed after approximately four weeks of incubation under methane (*Oshkin et al., 2014*). Only one microcosm (microcosm 150C 10) was dominated by non-methylotroph species. Specifically, a *Janthinobacterium* and a *Flavobacterium* species were present at highest relative abundances in this microcosm. These species were also noted as highly abundant in some of the previously characterized samples,

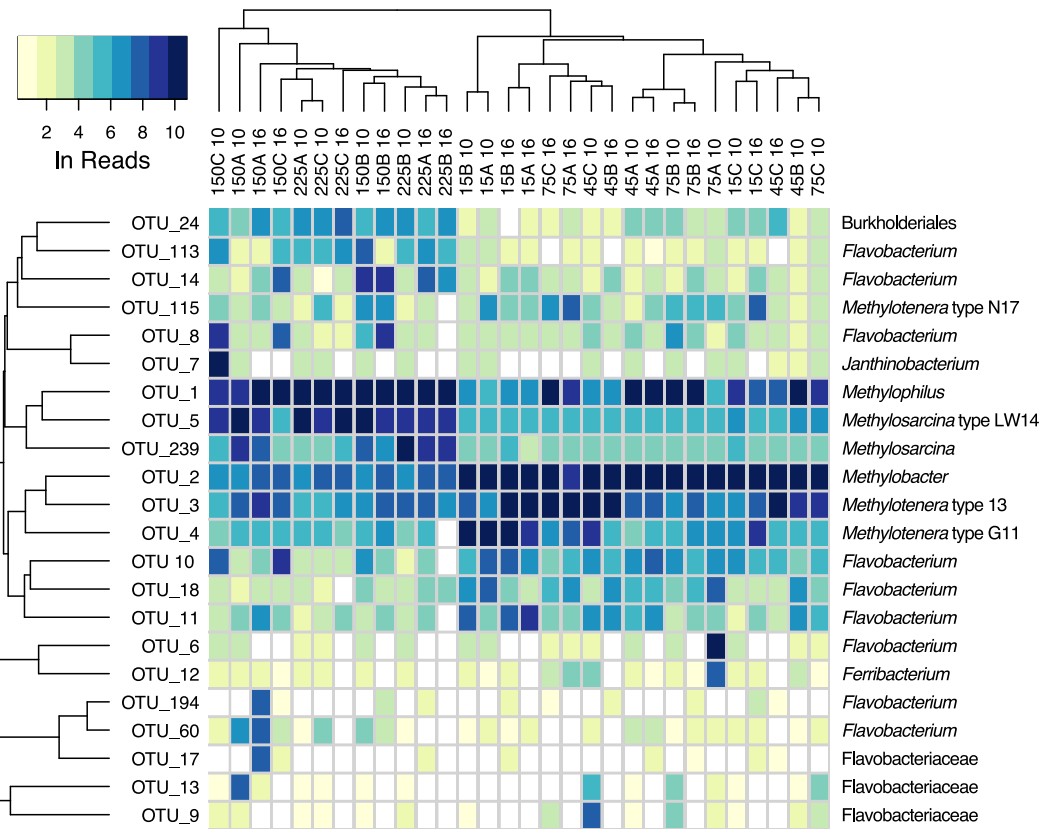

**Figure 3 Heatmap of major OTU relative abundances across samples.** Abundances were measured as Ln of reads. Sample designations are the same as in Fig. 2. Samples and OTUs were clustered hierarchically (average linkage), based on Bray-Curtis dissimilarity index of relative abundance profiles.

likely a result of a stochastic event of 'community crash' resulting in death and lysis of the dominant species (*Oshkin et al., 2014*).

Of the methanotroph types, a total of three OTUs were recognized: OTU_2 was classified as *Methylobacter*, and OTU_5 and OTU_239 were classified as *Methylosarcina* (Table 1). Of the *Methylophilaceae* types, a total of four OTUs were recognized, one classified as *Methylophilus* (OTU_1) and three classified as *Methylotenera* (OTU_3, OTU_4 and OTU_115). These were most closely related to, respectively, *Methylotenera mobilis* 13, *Methylotenera* sp. G11, and *Methylotenera* sp. N17 (Table 1), all isolated from Lake Washington. Most of the remaining persistent OTUs (more that 1% of total sequences in at least one sample) belonged to *Flavobacteriaceae* (Fig. 3 and Table S1).

In most of the microcosms (73.3%), the methanotroph types made up less than 50% of the total population, and in most (63.3%), the *Methylophilaceae* types were most relatively abundant (Fig. 2). These data support our prior observations on a strong response by *Methylophilaceae* to the methane stimulus, and on a successful carbon transfer between the methanotrophs and non-methanotrophs (*Oshkin et al., 2014*).

Hernandez et al. (2015), *PeerJ*, DOI 10.7717/peerj.801

**Table 1 Methylotroph OTUs, cultivated proxy organisms, and nitrate metabolism functions.** *Methylobacter* isolates from Lake Washington have not been formally described. Phenotypically and genomically they are similar to each other and to the described strain of *Methylobacter tundripaludum* (*Wartiainen et al., 2006*).

| OTU | Cultivated proxy organism | % 16S rRNA identity | Respiratory nitrate reductase | Respiratory nitrite reductase | Nitric oxide reductase | Nitrous oxide reductase | N$_2$ fixation machinery | Rnf complex | Hydrogenase |
|---|---|---|---|---|---|---|---|---|---|
| OTU_2 | *Methylobacter* 21/22[a] | 99.4 | + | + | − | − | + | + | + |
| OTU_2 | *Methylobacter* 31/32[a] | 99.4 | + | + | − | − | + | + | + |
| OTU_5 | *Methylosarcina lacus* LW14[b] | 99.8 | − | − | +[e] | − | − | − | − |
| OTU_3 | *Methylotenera* 13[c] | 99.8 | + | + | + | + | − | + | − |
| OTU_4 | *Methylotenera* G11[d] | 98.6 | − | + | + | − | − | + | − |
| OTU_115 | *Methylotenera* N17[d] | 99.6 | − | − | − | − | − | + | − |
| OTU_1 | *Methylophilus* 1[c] | 99.8 | − | − | +[e] | − | − | − | − |
| OTU_1 | *Methylophilus* Q8[d] | 99.8 | − | − | − | − | − | − | − |

**Notes.**

[a] Data from the IMG/JGI public database.

[b] Data from *Kalyuzhnaya et al. (2005)*.

[c] Data from *Beck et al. (2014)*.

[d] Data from *McTaggart et al. (2015)*.

[e] Gene product is likely nonfunctional.

We observed a dramatic difference between community responses to high (150 to 225 μM) versus low (15 to 75 μM) initial oxygen tensions, especially in terms of the major methane-oxidizing types. While the *Methylosarcina* types were dominant in high-oxygen microcosms, they were almost absent from the low-oxygen microcosms. Conversely, the *Methylobacter* types were dominant in the low-oxygen microcosms while constituting only a minor population in the high-oxygen microcosms (Figs. 2 and 3). We have not identified *Methylosarcina* species in our prior experiments, except for the native lake sediment communities, at low relative abundances (*Oshkin et al., 2014*). This is likely due to the fact that, in the previous study, the 'high' oxygen microcosms were only fed oxygen weekly, thus becoming hypoxic for a significant duration of time (*Oshkin et al., 2014*). This suggests that the *Methylosarcina* species are only competitive when oxygen is present, and that they become outcompeted by the *Methylobacter* types during hypoxia. Although it is unlikely, this behavior could simply be explained by the differences in oxygen affinity, as all of the oxygen concentrations used in this study were well above the reported Km values for methanotrophs (*Joergensen, 1985*; *Dunfield et al., 1999*). More likely, the differences are due to the different metabolic strategies employed during hypoxia (see below).

The occurrence of specific *Methylophilaceae* types was also oxygen-dependent. The *Methylophilus* types prevailed at higher oxygen tensions, and the *Methylotenera* types prevailed at lower oxygen tensions. Of the latter, OTU_3 was the most relatively abundant among the samples, and OTU_115 was the least relatively abundant (Figs. 2 and 3). However, the transition between the *Methylophilus* and *Methylotenera* types was more gradual. While the *Methylophilus* types were dominant at high oxygen tensions, they were present at variable levels at the intermediate oxygen tensions. The *Methylotenera* types were more represented in the samples with the lowest oxygen, suggesting competitive advantage for these species during hypoxia. These data are in agreement with our prior data on *Methylophilus* being more competitive in the conditions of higher oxygen and stable in the conditions of lower oxygen when no competitor is present (*Oshkin et al., 2014*).

The distribution of the non-methylotrophic heterotrophic species among the communities investigated was also nonrandom. As with the methylotrophs, a clear switch was observed between some of the prevailing satellite species between the high- and the low oxygen tension conditions. While the more prominent satellites in the high oxygen samples were OTU_14 (*Flavobacterium*), OTU_24 (*Burkholderiales*) and OTU_113 (*Flavobacterium*), the most prominent satellites in the low oxygen conditions were OTU_10, OTU_11 and OTU_18, all *Flavobacterium* species.

At this moment, we have no mechanistic knowledge of interactions within these methane-oxidizing communities, beyond the observation that carbon from methane does get transferred to *Methylophilaceae* and potentially to a broader range of microbes, based on stable isotope analysis (*Kalyuzhnaya et al., 2008*; *Beck et al., 2013*) and based on rapid population growth of *Methylophilaceae* and of certain non-methylotroph heterotrophs in the microcosms. However, the associations of methanotrophs with non-methanotrophs are persistent, and they do select for specific types. The communities are roughly stable

over time, with the methanotroph population typically oscillating between one third and two thirds of the community.

We carried out comparative genomics in order to obtain hints regarding which metabolic features might be responsible for oxygen level adaptation, including survival and/or growth during the periods of hypoxia. The genomes of two cultivated *Methylobacter* strains were compared to the genome of a *Methylosarcina* strain, and the genomes of *Methylotenera* strains were compared to the genomes of *Methylophilus* (see Table 1 for the list of organisms). Only a few metabolic features were uncovered that differentiated the functional counterparts, the most notable being nitrogen metabolism functions. The *Methylosarcina* genome only encoded functions for nitrate conversion into ammonium (assimilatory denitrification) and for a single, likely nonfunctional subunit of nitric oxide reductase. On the contrary, the *Methylobacter* genomes encoded, in addition, respiratory nitrate and nitrite reductases (Table 1). The *Methylobacter* genomes also contained genes predicted to encode functions essential to dinitrogen fixation, including the subunits of the Rnf complex that is essential for this metabolism, at least in some species (*Schmehl et al., 1993*). These genomes also encoded multiple hydrogenases and accessory functions. While at this moment the potential role of dinitrogen fixation in the fitness of *Methylobacter* is not obvious, its ability to denitrify presents a mechanism by which it may be able to out-compete *Methylosarcina* during hypoxia. Methanotrophy has been recently demonstrated during hypoxia, linked to nitrate reduction, in a related methanotroph (*Kits, Klotz & Stein, 2015*). Interestingly, one other difference between *Methylobacter* and *Methylosarcina* genomes was the presence of the *pxmABC* gene cluster (in the former but not the latter), encoding homologs of the subunits of methane monooxygenase (*Tavormina et al., 2011*). While the function of these genes remains unknown, they were found overexpressed during hypoxia in a denitrifying methanotroph (*Kits, Klotz & Stein, 2015*).

Likewise, while the *Methylophilus* genomes only encoded assimilatory denitrification reactions, the *Methylotenera* genomes varied in terms of their denitrification potential, from assimilatory in strain N17 to partial dissimilatory in strain G11 to complete dissimilatory in strain 13 (*Beck et al., 2014*). The denitrification capability has been experimentally demonstrated in at least one *Methylotenera* species (*Mustakhimov et al., 2013*). The *Methylotenera* genomes also encoded the Rnf complex, in the absence of any dinitrogen fixation genes.

It is tempting to speculate that nitrogen metabolism functions, and especially the denitrification capability, confer competitive advantage at low oxygen to both *Methylobacter* and *Methylotenera*. It is also possible that these organisms may exchange nitrogen species such as nitrite, nitric or nitrous oxide. However, as yet we do not have information regarding how a methanotroph can provide carbon to a community of non-methanotrophs and as to what advantage the methanotroph may be gaining from the satellite community.

Overall, the experiments described here provide further support to our observations on a special relationship between the *Methylococcaceae* and the *Methylophilaceae*, and also provide further support to the observation that non-methylotrophic species, especially

*Flavobacteriaceae*, may also play a role in this proposed mutualistic relationship. Moreover, we now conclude that oxygen availability is a major factor determining what species engage in cooperative behavior. At high oxygen tensions, *Methylosarcina* appear to have advantage over *Methylobacter*, and *Methylophilus* appears to have advantage over *Methylotenera*. At intermediate oxygen tensions, *Methylobacter* appears to cooperate with either *Methylophilus* or *Methylotenera*. At low oxygen tensions, including extended periods of hypoxia, *Methylobacter* and *Methylotenera* outcompete, respectively, *Methylosarcina* and *Methylophilus*. However, different types within each genus are also identifiable (see also *Oshkin et al., 2014*), and these may be selected by more discrete factors. These details will be addressed in future studies.

### Funding

This material is based upon work supported by the U.S. Department of Energy, Office of Science, Office of Biological and Environmental Research under Award Number DE-SC-0010556. The Mexican National Council for Science and Technology (CONACYT) provided Scholarship No. 208120 for Maria E. Hernandez. The funders had no role in study design, data collection and analysis, decision to publish, or preparation of the manuscript.

### Grant Disclosures

The following grant information was disclosed by the authors:
U.S. Department of Energy, Office of Science, Office of Biological and Environmental Research: DE-SC-0010556.
University of Washington eScience Institute.
The Mexican National Council for Science and Technology (CONACYT): 208120.

### Competing Interests

Ludmila Chistoserdova is an Academic Editor for Peer J.

### Author Contributions

- Maria E. Hernandez conceived and designed the experiments, performed the experiments, analyzed the data, wrote the paper, prepared figures and/or tables, reviewed drafts of the paper.
- David A.C. Beck analyzed the data, contributed reagents/materials/analysis tools, wrote the paper, prepared figures and/or tables, reviewed drafts of the paper.
- Mary E. Lidstrom wrote the paper, reviewed drafts of the paper.
- Ludmila Chistoserdova conceived and designed the experiments, analyzed the data, wrote the paper, prepared figures and/or tables, reviewed drafts of the paper.

### DNA Deposition

The following information was supplied regarding the deposition of DNA sequences:
NCBI Bioproject number PRJNA274703.

## Supplemental Information

Supplemental information for this article can be found online at http://dx.doi.org/10.7717/peerj.801#supplemental-information.

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
