# Peer review of "Oxygen availability is a major factor in determining the composition of microbial communities involved in methane oxidation"

_PeerJ, doi:10.7717/peerj.801_

## Round 0.1 · original submission · Minor Revisions

· Academic Editor

Minor Revisions

Both reviewers found a good deal of merit in your paper. However, in their comments they have both highlighted the need for further clarification of the results from the experiments and discussion in terms of the affinity of methanotrophs for oxygen, and Reviewer 2 has reasonably suggested that cycling times between oxic and anoxic conditions may have affected these results.

I also agree with Reviewer 2's suggestion that more information on the extent of the bioinformatics (methodologies?) and specific data sources (with references) used for the comparative genomic analysis are needed to justify the preliminary conclusions.

Therefore, in addition to addressing all of the reviewers' individual comments in the usual way, I would be very grateful if you would give careful consideration to these two issues.

·

Basic reporting

This current study aims to address the impact of oxygen tension on methanotroph and methylotroph communities in a lake sediment using microcosm set-ups. The manuscript is very well written and data is clearly presented.

Experimental design

I felt the information presented in supplementary figure 1 is important for the readers to fully appreciate the experimental design and should therefore be included in the main manuscript, considering the study is relatively short.

Furthermore, it would be useful if the authors could explain the unusual methane:oxygen consumption ratio observed in these bottle incubations. For example, when the O2 tension was 15 uM, 0.5 mM O2 was consumed while ~ 2mM CH4 was depleted. Even if only methanotrophs consumes all O2, theoretically only 0.25 mM CH4 can be oxidized. Does this imply that O2-independent methane oxidation has occurred in these incubations? Please comment.

At low O2 tensions, O2 levels dropped significantly after 8 hours and thereafter the microorganisms in the incubations experienced severe O2 limitation. Is it therefore possible that the changes of microbial community observed (e.g. data shown in figure 1) reflects different strategies of M bacter vs M sarcina & M tenera vs M philus in dealing with hypoxia? The authors need to rule out for example, Methyloobacter (and Methylotenera) simply survives better at microaerophilic or anaerobic conditions. If this is indeed the case, please consider revise the title as it may be misleading.

It would be useful if the authors could comment on whether M bacter and M sarcina may have different apparent affinity towards O2 or methane? As the authors discussed in the manuscript, if oxygen tension selects for methanotrophs (and methylotrophs) populations, one would expect that Mbacter may have higher affinity to O2 or CH4 than Msarcina. With cultures available, it would be informative to carry out such tests. Please discuss.

Validity of the findings

N/A

Comments for the author

I felt the data presented in table 1 and subsequent discussion with regard to the present/absence of certain genes of interest is difficult to follow. what is the take home message of these comparative genomics data? for example, the speculation of N-metabolism may confer competitive advantage at low oxygen to Mbacter and Mtenera. But why? How is this relevant to the study of the impact of O2 tension on CH4 oxidation. Please consider revise to make it easier to follow.

Reviewer 2 ·

Basic reporting

This is a well written and nicely presented paper and I have only a few minor comments.

Experimental design

The findings appear to be largely confirmatory of an earlier paper by Oshkin et al, 2014. The Introduction (line 61) states only that "additional experimental support is necessary". It should more clearly explain how the present study differs in its design from the previous study and what the results were expected to add to the previous findings. The concluding paragraphs should also clearly delineate what the present study has contributed.

line 225 Some discussion of the affinity of methanotrophs to O2 would be useful. All of the initial O2 concentrations are above the Km of the monooxygenase for O2, so the key to the differences in the treatments is probably the times of cycling between oxic and anoxic conditions in each vial- i.e. how long each is anoxic before the atmosphere is reconstituted.

line 203 ff Please provide some details of how the genomes were compared. Are the comparisons purely intuitive, or were reciprocal searches performed to determine all genes occurring in only one genome? Some Methylobacter contain a second pmo operon of unknown function- do these isolates? Please also provide a description of the isolates used for genome comparisons. I assume these are isolates obtained from previous studies of Lake Washington? Please provide references.

line 101ff How were representative sequences of the OTUs selected? Were they consensus sequences, or the most frequently occurring sequence, or were they randomly selected?

Validity of the findings

line 198-202 This argument, citing unpublished data, is not convincing and should be removed. One would need much more detail to judge this claim.

Comments for the author

line 22 "effects response by" poor grammar
line 43 Methane need not be the sole source of carbon. Some methanotrophs are autotrophic.
line 52 "mutualism" is perhaps too strong a term. There is certainly some cross-feeding, but we do not yet know if the methanotroph somehow benefits from the partnership.
line 61 "in order to shift the accepted paradigms" This wording is too strong. I think it is already well accepted that methanotrophs support some cross-feeding with other bacteria.
line 72 what was the flow rate?
line 91-92 I do not understand this statement. What samples were pooled, and what are "equal proportions based on molecular weight and DNA concentrations"?
line 96 "Sequences were depleted of barcodes". Change to "Barcodes were removed"
line 143 What is a possible explanation for the predominance of other non-methylotrophic organisms in one microcosm?

---

## Round 0.2 · accepted · Accept

· Academic Editor

Accept

Thank you for responding to all of the reviewers' points and making appropriate modifications to you paper, including the title change, which I consider to have satisfactorily addressed the substantive issues raised in review.